# Impacts of the COVID-19 Pandemic on Wildlife in Huangshan Scenic Area, Anhui Province, China

**DOI:** 10.3390/ani15060857

**Published:** 2025-03-17

**Authors:** Yuting Lu, Yaqiong Wan, Lanrong Wang, Dapeng Pang, Yinfan Cai, Yijun Wu, Mingxia Tang, Jiaqi Li, Baowei Zhang

**Affiliations:** 1School of Life Sciences, Anhui University, Hefei 230601, China; kexingluovo@163.com (Y.L.); d22201015@stu.ahu.edu.cn (L.W.); pia@163.com (D.P.); yinfancai@gmail.com (Y.C.); 2Key Laboratory of Biodiversity and Biosafety, Nanjing Institute of Environmental Sciences, Ministry of Ecology and Environment, Nanjing 210042, China; wanyaqiong0229@163.com; 3Bureau of Park and Wood of Huangshan Scenic Area Management Committee, Huangshan 245800, China; wuyijun28@163.com (Y.W.); tmx4414@163.com (M.T.)

**Keywords:** wildlife disturbance, species population trends, habitat use, anthropogenic disturbances, wildlife survival, coronavirus disease 19

## Abstract

Huangshan, a famous mountainous scenic area and biodiversity hotspot in East China, is particularly vulnerable to human activities and habitat fragmentation, more so than designated nature reserves. The COVID-19 pandemic has led to significant global shifts in human activity, providing an unprecedented opportunity to study the impact of anthropogenic disturbances on wildlife survival. Camera data from before and during the pandemic were analyzed to explore the changes in population size, habitat use, and temporal activity of the local species. This study highlights the negative impacts of human activity on wildlife, providing essential data to support conservation and management in the Huangshan Scenic Area.

## 1. Introduction

The Huangshan Scenic Area, a global geopark and biosphere reserve, is a biodiversity hotspot in East China with high forest coverage and complex terrain, providing a suitable habitat for the survival and reproduction of various rare plants and animals [1,2,3]. However, unlike nature reserves that focus on ecological conservation, scenic areas experience high-intensity human interference, with a management model primarily geared toward economic development rather than ecological protection. Therefore, scenic areas are more vulnerable to anthropogenic disturbances and habitat fragmentation than nature reserves, and local wildlife may also face greater threats [4]. Substantial evidence indicates that ecotourism significantly affects species’ reproduction and survival, especially in isolated or disturbance-sensitive populations [5,6]. As the first tourist destination in China to be included in the Man and the Biosphere (MAB) Program, the Huangshan Scenic Area uniquely combines “high-intensity tourism” with “high biodiversity”, making it an ideal model for testing the theory of human–wildlife coexistence [7].

Human activities threaten the stability and diversity of ecosystems [8,9], contributing to issues such as global warming, urbanization expansion, and species extinction [10]. High human disturbance impacts local wildlife by creating “landscapes of fear” [11,12], forcing animals to adjust their behaviors to avoid humans in both time and space [13,14], potentially leading to non-lethal physiological and fitness impacts [15]. However, the indirect, non-lethal pathways through which humans alter ecosystems have been largely underexplored.

The global onslaught of the COVID-19 pandemic produced an immense challenge for humanity and greatly impacted public health systems [16,17]. Moreover, it temporarily halted the Anthropocene’s expansion and led to shifts in population mobility, known as the “Anthropause” [18,19], offering a unique opportunity to study wildlife responses to reduced human activity [20,21].

Early investigations indicated that the outbreak immediately reduced human activities in industry and tourism [22,23], resulting in a series of positive ecological effects [24,25], such as decreased water and air contamination [26,27]. In addition, 275 species experienced population shifts or occupied unusual areas owing to reduced human mobility [28]. The pandemic also appeared to increase daily activity in nocturnal/crepuscular species [29]. Reports frequently mentioned unusual wildlife sightings in urban areas, including wolves (*Canis lupus*) and deer, suggesting that wildlife altered their activity patterns and enhanced their utilization of the surrounding habitats during the pandemic [30]. However, the decline in law enforcement has fostered opportunities for poachers to illegally hunt wildlife, negatively impacting wildlife and conservation efforts [31,32]. Additionally, there is evidence indicating that various wild and domesticated animals are susceptible to SARS-CoV-2 [33,34]. Overall, the pandemic has had complex positive and negative impacts, potentially inducing chain reactions that affect wildlife and nature conservation [35,36,37].

The Huangshan Scenic Area experienced a dramatic change in human activity as a popular tourist destination before and during the pandemic. Official data demonstrated that the flow of visitors to the scenic area averaged 3.42 million per year during the 3 years before the outbreak and then dropped to an average of 1.53 million per year during the 3 years after the outbreak (https://hsgwh.huangshan.gov.cn/, accessed on 16 January 2025). To understand how this disruption affects wildlife, it is essential to examine the impact of anthropogenic habitat factors on species distribution and behavior [38]. To explore the extent and scale of this impact, we conducted surveys of large- and medium-sized mammals and ground-dwelling birds in the Huangshan Scenic Area based on camera trapping across several years of substantial changes in human disturbance. With the advantages of continuous monitoring and non-invasiveness [39,40], camera trapping is an effective tool for researching wildlife populations and habitats, providing crucial information on diversity, distribution, behavior, and activity patterns, especially for elusive or nocturnal species [41,42].

This study aimed to compare the anthropogenic effects before and during the COVID-19 pandemic on the population size, habitat use, and diurnal activity of seven wild species with the highest relative abundance indices (*Muntiacus reevesi*, *Lophura nycthemera*, *Macaca thibetana*, *Paguma larvata*, *Capricornis sumatraensis*, *Sus scrofa*, and *Arctonyx collaris*) in the Huangshan Scenic Area, China.

## 2. Materials and Methods

### 2.1. Study Area

The Huangshan Scenic Area (118°01′–118°17′, 30°01′–30°18′) is located in Anhui Province in central China (Figure 1), covering an area of 160.6 km^2^ and divided into six management zones. It is bordered by five towns and a forest farm, and the highest peak reaches 1864 m. This scenic area has a subtropical monsoon climate, with an average annual precipitation of 1670 mm, an average annual temperature ranging from 20 to 40 °C, and a frost-free period of 220 d. The local forest cover reaches 98.29%, with the vegetation mainly comprising subtropical rainforests and subtropical evergreen broad-leaved forests [43,44].

Tourism is the primary human activity, alongside limited resource gathering by local residents. Domesticated animals (cats and dogs) are the only animals present, with no grazing observed.

### 2.2. Data Collection

Sixty cameras, models Ltl 5210 MC and Ltl 6210 MC (Ltl Acorn Co., Ltd., Zhuhai, China), were deployed in the Huangshan Scenic Area from March 2017 to December 2022 (Figure 1). To analyze the coexistence between humans and wildlife, cameras were deployed near roads or tourist facilities, with concealed water sources or animal trails selected as placement sites to maximize wildlife detection. They were positioned at least 0.4 km apart horizontally and distributed along an elevation gradient of 381–1729 m, with 2–15 cameras placed within every 200 m elevation range. Additionally, camera sites were proportionally allocated across every vegetation type, including evergreen broadleaf forests (26 sites), evergreen coniferous forests (18 sites), deciduous broadleaf forests (13 sites), and shrubs (3 sites). Verification confirmed that human activity was detected at 52 sites during this period.

We selected open and front-lit points to reduce false trigger effects [45], then deployed the cameras at a height of 0.3–0.6 m above the ground. The latitude and longitude coordinates, vegetation, and altitude of each camera trap station were recorded. The cameras operated 24 h a day, programmed to take three photos upon triggering, followed by a 10-s video, with a delay of at least 1 min between consecutive events. Batteries and SD cards were replaced and collected every 6 months.

The epidemic’s widespread impact starting in February 2020 caused a significant drop in visitor numbers, with March 2017 to January 2020 defined as the ‘before pandemic (BP)’ period, and February 2020 to December 2022 as the ‘during pandemic (DP)’ period, both lasting 35 months.

### 2.3. Data Analysis

#### 2.3.1. Relative Abundance Index

Image records were screened to identify ‘independent photographs’ according to a standard temporal separation criterion of more than 30 min between consecutive images of the same species to avoid repeated counting of a single individual during a transitory stay close to the camera trap [46,47]. The relative abundance index (RAI) was calculated to evaluate the relative population sizes of bird and mammalian species using the following formula:(1)$$\mathrm{RAI}=\frac{\sum_{i=1}N_{i}}{\sum_{i=1}T_{i}}\times 100$$
where *Ni* is the number of independent photographs of the *i*-th species, and *Ti* is the total number of camera trapping days [48]. Monthly RAI was analyzed to visualize population changes over time, and the Wilcoxon Signed-Rank test was used to assess significant differences in relative abundance between the two periods for the seven species at each camera site [49].

#### 2.3.2. Occupancy and Detection Probability

For the final year of each period (2019 and 2022), a single-season occupancy model was constructed for the period with the highest number of independent photographs (May–July) to estimate occupancy (*ψ*) and detection (*p*) probabilities and explore the effects of relevant environmental factors and human activities [50]. In this study, the habitat use of individual species was assumed to be independent of the others, and the data from each camera trap were considered to be repeated observations of an independent station. The 3-month detection history for each species at 60 camera sites was established, using “1”, “0”, and “NA” to represent the situations of “detected”, “undetected”, and “camera malfunction”, respectively [51]. To determine the relationship between habitat features and wildlife occupancy, several anthropogenic and habitat covariates that could potentially affect wildlife activity were measured [52,53], including elevation (ELE), normalized vegetation index (NDVI), domestic animal RAI (DRAI), human RAI (PRAI), and distance from the nearest road (DNR) and tourist facilities (DNT). All site covariates were ensured to be non-significantly correlated using Spearman correlation analysis and were standardized to Z-scores before modeling [54]. Models were constructed using the “unmarked” package in R version 4.3.3 (Vienna, Austria, accessed on 22 May 2024) [55,56], following the stepwise model selection procedures described by Burnham and Anderson [57], where candidate models with the lowest Akaike information criterion (AIC) values were considered the best descriptors of species occupancy and detection probability.

#### 2.3.3. Activity Pattern Overlap

The overlap coefficients (Δ) of focal species activity under two scenarios were calculated using the kernel density estimation [58], and clock-recorded times were converted to solar time before analysis [59]. The temporal overlap analysis was conducted using the “overlap” package in R version 4.3.3 (Vienna, Austria) with the estimator ∆_4_, since the sample size of all surveyed species was much larger than 75 [60]. *p*-values for estimated coefficients were derived using an approximation of the Wald statistic, defined as the coefficient estimate divided by its standard error, with *p* < 0.05 considered significant for all statistical tests [61]. In addition, the nocturnality was quantified using the night-time relative abundance index (NRAI) [62], which is the percentage of nocturnal detections. Subsequently, the risk ratio (RR) was calculated for each species, reflecting the comparative nocturnality shift in different periods [63], using the following formula:(2)$$\mathrm{RR}=\ln\left(\frac{{\mathrm{NRAI}}_{\mathrm{BP}}}{{\mathrm{NRAI}}_{\mathrm{DP}}}\right)$$
where *NRAI_BP_* is the night-time relative abundance index before the pandemic, and *NRAI_DP_* is the index during the pandemic.

## 3. Results

In this study, 60 camera sites were surveyed from 2017 to 2022, with 95,523 camera trapping days, resulting in a total of 20,866 independent photographs of identifiable wildlife, as well as 843 independent detections of humans and 557 of domesticated animals (cats and dogs). Seven focal species with the highest relative abundance of mammals (except rodents) and ground-dwelling birds were selected for further study: Reeves’ muntjac (*M. reevesi*), silver pheasant (*L. nycthemera*), Tibetan macaque (*M. thibetana*), masked palm civet (*P. larvata*), mainland serow (*C. sumatraensis*), wild boar (*S. scrofa*), and hog badger (*A. collaris*).

### 3.1. RAI

Trends in the monthly relative abundance of these seven species over 6 years were analyzed, with humans and domestic animals similarly included in the statistics. The results showed that the peak activity of most focal species occurred during the pandemic (Figure 2).

As shown in the comparison of the two scenarios, the relative abundances of humans (*p* = 0.025) and domesticated animals (*p* = 0.002) were significantly lower during the pandemic, confirming previous hypotheses regarding the impact of the outbreak on human activities. Consistent with our prediction, the relative abundances of Reeves’ muntjac, silver pheasant, and wild boar (*p* < 0.01) were significantly higher during the pandemic than before, while masked palm civet and hog badger also showed a moderate increase in population size (*p* < 0.05). This suggests a general trend of rapid population growth among most focal species, likely driven by the substantial reduction in human activity during the pandemic. However, no significant differences were observed for Tibetan macaque or mainland serow (*p* > 0.05), indicating that their population sizes remained unaffected by changes in human disturbance intensity. This may suggest that these species either exhibit excessive habituation to human activity or heightened avoidance of it (Figure 3).

### 3.2. Habitat Use

Separate occupancy models for the seven species during both periods were constructed, using combinations of environmental variables that were not significantly correlated (*p* > 0.05, Figure A1), with the model yielding the lowest AIC value selected for analysis. Four species showed a substantial increase in naïve occupancy after the outbreak, with the masked palm civet showing the largest increase, from 0.35 to 0.74 over the three-year period. Similar changes were also observed for the hog badger (0.34; 0.71), Reeves’ muntjac (0.40; 0.72), and silver pheasant (0.46; 0.62). In addition, the naïve occupancy of Tibetan macaques (0.55; 0.57), mainland serows (0.26; 0.29), and wild boars (0.21; 0.28) showed a slight upward trend before and during the pandemic. Overall, all focal species in the study showed higher naïve occupancy rates during the pandemic than before, indicating that reduced human disturbance led to increased habitat use and an expanded distribution range (Table 1).

Based on the results of the occupancy model, the detection rates for all species were higher during the outbreak than before, and the shift trends in occupancy rates were generally similar to those of naïve occupancy, except for mainland serow, whose occupancy rate decreased slightly during the pandemic (Table 2).

The covariate analysis results indicated that elevation was the most influential factor affecting the occupancy probability of Reeves’ muntjac and silver pheasant, both of which tended to be distributed in low-altitude areas. No significant correlations were observed for the occupancy probability of other species.

The detection probability of all focal species exhibited either direct or indirect correlations with human disturbance: two species showed a direct negative correlation with human and domestic animal activity. Additionally, three species avoided tourism facilities, and two species avoided roads, indirectly reflecting an association with human disturbance. The detection probability of Reeves’ muntjac and silver pheasant at independent sites was significantly and negatively correlated with the relative abundance of humans (*β* = −5.97, −2.10) and domestic animals (*β* = −2.09, −1.44), with this strong correlation appearing to weaken during the pandemic. The distance from the nearest tourist facilities was an important factor for the detection probability of mainland serow (*β* = 1.01), wild boar (*β* = 0.72), and masked palm civet (*β* = 0.88). Hog badgers were more frequently detected at camera sites farther from roads (*β* = 0.93) and with higher domestic animal activity (*β* = 0.28). Tibetan macaque had a tendency to occupy low elevation areas away from roads in both scenarios but showed no direct correlation with the presence of humans and domestic animals (Table 3).

### 3.3. Temporal Overlap

Kernel density estimation revealed that human disturbances primarily occurred during the daytime, while domestic animals impacted wildlife throughout the day. Humans reduced midday activity during the pandemic, with peaks shifting toward the early morning, while domestic animals intensified their nocturnal activity frequency, both showing significant changes in activity patterns between the two scenarios (*p* < 0.01).

Diurnal species such as Reeves’ muntjac, silver pheasant, Tibetan macaque, and wild boar exhibited similar activity peaks at dawn (6:00–8:00) and dusk (16:00–18:00), with the Tibetan macaque being mainly active at midday (13:00). The activity peaks of nocturnal species were mainly concentrated at 20:00 and 24:00. The Reeves’ muntjac (*p* < 0.01), silver pheasant (*p* = 0.01), and masked palm civet (*p* = 0.04) displayed significant shifts in activity patterns between the pre-pandemic and pandemic periods. In addition, a noteworthy phenomenon was observed in that most species (except Tibetan macaques and wild boars) showed attenuation of nocturnal behavior during the pandemic (RR > 0). This suggests that the decrease in human activity alleviated daytime pressure, leading to a shift in the species’ temporal niches (Figure 4).

## 4. Discussion

The COVID-19 pandemic has changed the intensity and scope of human activity in an unprecedented way, leading to a series of positive impacts [64,65]. Previous studies have shown that wildlife occupied new areas or altered their abundance during the pandemic [18,66]. However, the reduction in law enforcement potentially has exposed wild animals to an increased risk of poaching [67].

As predicted, human activity in the Huangshan Scenic Area changed dramatically before and during the pandemic, with a significant decline in humans and domestic animals, reflecting global anthropogenic trends [68,69]. In addition, camera analysis showed a significant drop in travelers but a slight rise in resource gathering and poaching [70]. This shift altered activity rhythms, with human activity peaking in the morning and decreasing at midday, while the nocturnal activity of domestic animals increased.

The global expansion of human activities has significantly affected wildlife [8], and the fear of humans has forced animals to limit their range in order to avoid making contact [13,71]. However, not all species are equally affected, and traits such as broad habitat tolerance, nocturnality, and small body size may contribute to greater flexibility in response to human disturbance [72]. Our study showed that, except for the mainland serow, all focal species exhibited varying improvements in relative abundance and habitat use, indicating that reduced human activity during the pandemic eased wildlife fear, positively influencing survival and reproduction [28].

As the mammal and bird species with the highest relative abundance in the scenic area, the Reeves’ muntjac and the silver pheasant, respectively, showed similar changes before and during the pandemic. Their detection probability was significantly negatively correlated with the relative abundance of humans and domestic animals, suggesting avoidance of human disturbance [73]. However, their habitat selection did not show strong rejection of highly disturbed areas such as roads and tourist facilities. Thus, as human activity decreased, these species rapidly expanded into surrounding habitats, showing a notable increase in both population size and habitat range. Some species have become habituated through repeated exposure to humans, leading to a reduction in their behavioral responses [74,75]. In particular, in areas frequented by tourists, certain macaque species have become highly adapted to the presence of humans [76]. Consequently, the population size of Tibetan macaques remained relatively stable and even exhibited slight signs of decline during the pandemic. Populations deprived of provisions move closer to human communities in search of food [77], possibly explaining the slight rise in their occupancy probability.

Wild boar populations have gradually increased over the past few decades owing to their adaptability to environmental conditions [78,79]. Consequently, the increase in wild boar population and detection rates during the pandemic may have been triggered by a combination of adaptability and reduced human disturbance. However, their habitat use changed little, suggesting that the presence of tourist facilities, rather than human activity itself, was the primary factor influencing their habitat selection [80]. Other large ungulates that are sensitive to tourist facilities face similar situations. Mainland serows avoided both tourist facilities and human activities, and showed little expansion in population size or habitat use during the pandemic. In contrast, smaller carnivores, such as the masked palm civet and hog badger, saw significant increases in both numbers and habitat use. Although these species also avoided tourist areas and roads, they were less affected by human disturbances than larger animals, inferring that small body size may be a key trait for behavioral plasticity to human activity [81].

While most studies on wildlife response to human disturbance focus on spatial avoidance, the steady expansion of human activity is increasingly limiting available refuge for animals [9]. To cope with this expansion, some species alter their activity rhythms to minimize encounters with humans [14]. Despite a reduction in human activities during the pandemic, local wildlife showed minimal changes in their temporal patterns, with only the Reeves’ muntjac, silver pheasant, and masked palm civet showing significant differences, likely attributable to the large number of samples. In addition, the night-time relative abundance decreased across all species except for the Tibetan macaque and wild boar, with diurnal species exhibiting higher RR relative to nocturnal species. This suggests that wild animals are forced to adjust their temporal rhythms and shift their activities to night-time to avoid daytime human disturbances. Diurnal species appear more vulnerable to human impact, aligning with previous studies [29,63].

Although wildlife can coexist with humans by altering their activity rhythms to increase habitat use, this shift may have negative and far-reaching ecological consequences. Such behavioral changes can impose substantial and unnecessary health costs on wildlife [82], analogous to predation risk effects in predator–prey systems, in which costly antipredator behavior compromises prey reproduction and survival.

This study highlights the complex ecological impacts of human interference and confirms the pandemic’s positive effect on wildlife in high-traffic tourist areas. The findings support the designation of protected areas within scenic regions and inform the development of sustainable ecotourism policies, promoting a balanced approach to both ecological conservation and tourism development. For example, scenic areas could estimate their carrying capacity and establish new tourist routes to mitigate the negative impact of sightseeing activities on wildlife.

However, this study has certain limitations. Although no significant environmental changes were observed during the pandemic apart from reduced human activity, some species (such as wild boar) had already shown signs of population expansion in recent years. Therefore, their population changes cannot be entirely attributed to human disturbance. Additionally, since both human and domestic animal activities declined simultaneously during the pandemic, it was difficult to isolate the effects of each disturbance. Future research could adopt a spatial approach by linking wildlife detection at different camera sites to varying levels of disturbance, providing a more detailed understanding of wildlife responses to human activity.

## 5. Conclusions

The reduction in human disturbance during the COVID-19 pandemic period allowed most of the seven focal bird and mammal species to expand both their population sizes and habitat use, as well as increase their daytime activity. This suggests that anthropogenic disturbances imposed substantial stresses on the ecosystems before the outbreak, with diurnal species being more susceptible to the effects of human activities. This worldwide sanitary crisis has highlighted the intricate connections between humans, nature, and climate change, providing valuable scientific insights that can guide innovative strategies for wildlife–human coexistence.

## Figures and Tables

**Figure 1 animals-15-00857-f001:**
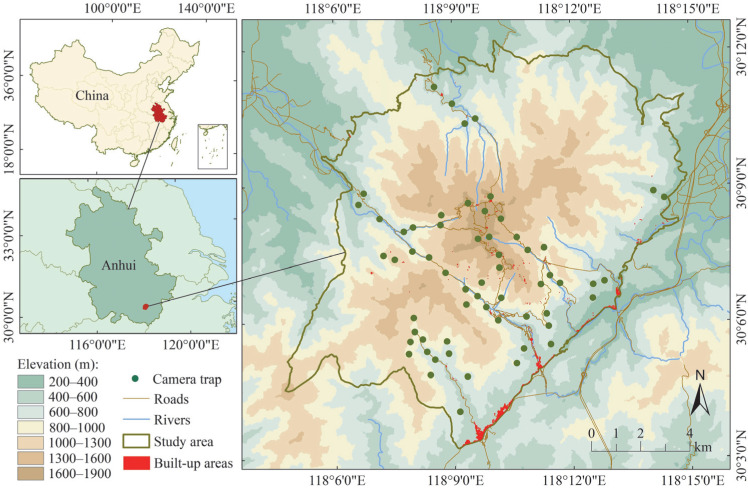
Camera trap site distributions in the Huangshan Scenic Area, China (Altitude and Remote Sensing Image map layer available online at www.gscloud.cn/, accessed on 8 September 2024).

**Figure 2 animals-15-00857-f002:**
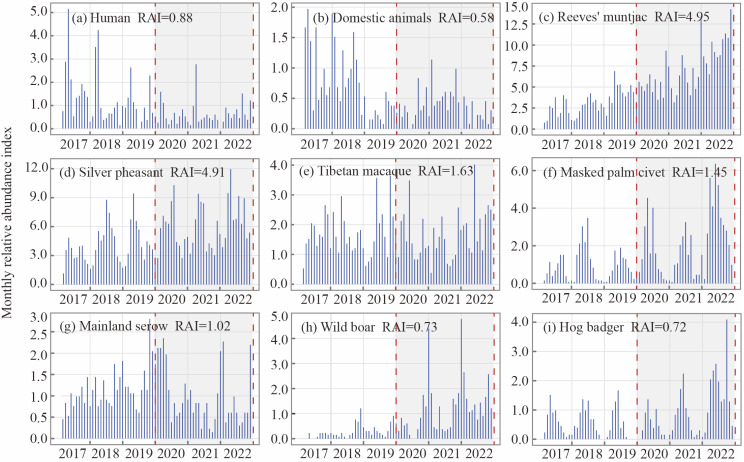
Monthly RAI and average RAI of focal species and human disturbance from 2017 to 2022.

**Figure 3 animals-15-00857-f003:**
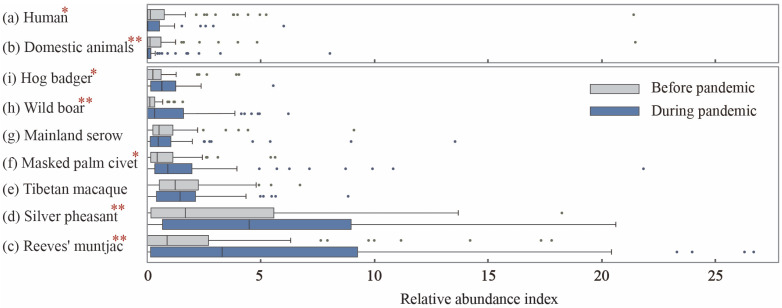
Differences in the relative abundance index (RAI) of human, domesticated animals, and focal species before and during the pandemic. * *p* < 0.05; ** *p* < 0.01.

**Figure 4 animals-15-00857-f004:**
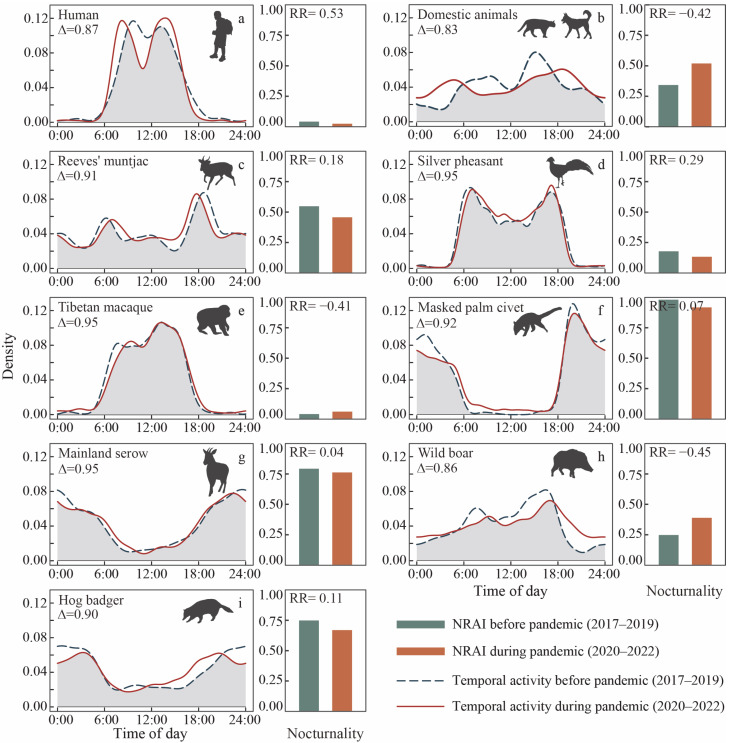
Activity pattern differences of humans (**a**), domestic animals (**b**), and the focal species surveyed in this study (**c**–**i**). For each species, the overlap (Δ) of temporal activity between the pre-pandemic (green dashed line) and pandemic (red solid line) periods is shown on the left, while the night-time relative abundance index (NRAI) and risk ratio (RR) between the pre-pandemic (green bar) and pandemic (red bar) periods are shown on the right.

**Table 1 animals-15-00857-t001:** Model selection parameters and naïve occupancy from the top-ranking model.

Species	Scenario	Best Model	Naïve Occupancy	AIC	nPars
(c) Reeves’ muntjac	BP	*Ψ* (ELE), *p* (PRAI, DRAI)	0.40	651.14	5
	DP	*Ψ* (ELE, PRAI), *p* (ELE, DRAI)	0.72	1022.47	6
(d) Silver pheasant	BP	*Ψ* (DRAI), *p* (ELE, NDVI)	0.46	1012.43	5
	DP	*Ψ* (ELE, NDVI), *p* (DNR, PRAI)	0.62	906.31	6
(e) Tibetan macaque	BP	*Ψ* (DNT), *p* (ELE, DNR)	0.55	562.94	5
	DP	*Ψ* (ELE, NDVI), *p* (ELE, DNR)	0.57	505.28	6
(f) Masked palm civet	BP	*Ψ* (PRAI, DRAI), *p* (NDVI, DNT)	0.35	345.51	6
	DP	*Ψ* (.), *p* (NDVI, DNT)	0.74	809.35	4
(g) Mainland serow	BP	*Ψ* (ELE, PRAI), *p* (NDVI, DNT)	0.26	207.1	6
	DP	*Ψ* (DNT, PRAI), *p* (NDVI, PRAI)	0.29	182.63	6
(h) Wild boar	BP	*Ψ* (PRAI, DRAI), *p* (ELE)	0.21	104.83	5
	DP	*Ψ* (NDVI, DRAI), *p* (ELE, DNT)	0.28	265.07	6
(i) Hog badger	BP	*Ψ* (PRAI, DRAI), *p* (NDVI, PRAI)	0.34	333.38	6
	DP	*Ψ* (DNR), *p* (DNR, DRAI)	0.71	544.38	5

Naïve occupancy: occupancy rate calculated without models; AIC: Akaike information criterion; nPars: number of model parameters; BP: before pandemic period; DP: during pandemic period; *Ψ*: probability of occupancy; *p*: probability of detection.

**Table 2 animals-15-00857-t002:** Occupancy and detection rates of focal species in two scenarios from the best model.

Species	Scenario	Occupancy		Detection	
		Occupancy Rate	SE	Detection Rate	SE
Reeves’ muntjac	BP	0.53	0.10	0.18	0.02
	DP	0.73	0.09	0.25	0.02
Silver pheasant	BP	0.55	0.10	0.16	0.02
	DP	0.64	0.22	0.25	0.02
Tibetan macaque	BP	0.58	0.08	0.10	0.01
	DP	0.67	0.12	0.12	0.02
Masked palm civet	BP	0.43	0.08	0.11	0.03
	DP	0.85	0.07	0.14	0.02
Mainland serow	BP	0.41	0.16	0.05	0.02
	DP	0.36	0.12	0.08	0.03
Wild boar	BP	0.32	0.09	0.02	0.01
	DP	0.36	0.13	0.08	0.02
Hog badger	BP	0.41	0.11	0.07	0.02
	DP	0.76	0.11	0.12	0.02

BP: before pandemic period; DP: during pandemic period.

**Table 3 animals-15-00857-t003:** Parameter estimates for explanatory variables in two scenarios from the best model.

Species	Scenario	Occupancy			Detection		
		Covariates	*β* _Estimates_	SE	Covariates	*β* _Estimates_	SE
Reeves’ muntjac	BP	ELE	−0.84 *	0.44	PRAI	−5.97 ***	1.44
					DRAI	−2.09 **	0.64
	DP	ELE	−1.04 *	0.42	ELE	−0.35 ***	0.11
		PRAI	1.38	2.23	DRAI	−0.02	0.13
Silver pheasant	BP	ELE	−0.83 *	0.39	PRAI	−2.10 **	0.77
					DRAI	−1.44 *	0.61
	DP	ELE	−1.34 *	0.47	DNR	−0.43 ***	0.12
		NDVI	−0.92 *	0.47	PRAI	−0.20 *	0.09
Tibetan macaque	BP	DNT	0.31	0.41	ELE	−0.58 *	0.25
					DNR	0.26 *	0.13
	DP	ELE	4.41	2.34	ELE	−0.97 ***	0.18
		NDVI	−2.73	1.58	DNR	0.84 ***	0.21
Masked palm civet	BP	PRAI	−4.82	3.56	NDVI	−0.36 *	0.15
		DRAI	10.88	8.93	DNT	0.88 ***	0.22
	DP				NDVI	−0.79 ***	0.11
					DNT	0.57 ***	0.09
Mainland serow	BP	ELE	−0.49	0.90	NDVI	−0.45 *	0.19
		PRAI	−7.05	8.18	DNT	1.01 ***	0.30
	DP	DNT	0.95	0.85	NDVI	−0.48	0.30
		PRAI	7.32	4.57	PRAI	−1.61 *	0.79
Wild boar	BP	PRAI	−36.39	137.10	ELE	−0.89	0.80
		DRAI	18.45	65.96			
	DP	NDVI	0.24	0.47	ELE	−1.04 *	0.43
		DRAI	0.80	0.95	DNT	0.72 ***	0.17
Hog badger	BP	PRAI	2.15	3.63	NDVI	−0.10	0.15
		DRAI	0.92	1.06	PRAI	−0.14	0.19
	DP	DNR	−1.41	0.79	DNR	0.93 ***	0.22
					DRAI	0.28 ***	0.07

BP, before the pandemic period; DP, during the pandemic period; ELE, elevation; NDVI, normalized vegetation index; DNR, distance from the nearest road; DNT, distance from the nearest tourist facilities; PRAI, people relative abundance index; DRAI, domestic animal relative abundance index. Significance codes: *p* < 0.05 *, *p* < 0.01 **, *p* < 0.001 ***.

## Data Availability

The original contributions presented in this study are included in the article. Further inquiries can be directed to the corresponding author.

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
