# Peer review of "Impacts of the COVID-19 Pandemic on Wildlife in Huangshan Scenic Area, Anhui Province, China"

_animals, 2025, doi:10.3390/ani15060857_

Round 1

Reviewer 1 Report

Comments and Suggestions for Authors

Dear authors,

this research article compares the activity and behavior of the 7 most relatively abundant species in the Huangshan Scenic area, before and after the beginning of the COVID-19 pandemic and the consequent reduction of human activity. Local wild animal population was monitored by using camera trapping.

I have to admit my limits in evaluating the methodologies of this work, since it's quite out of my competence field. Nevertheless, in my opinion the paper seems written properly, the results are described clearly and also the methods seem to me well presented, even if I cannot evaluate whether the statistics behind it is correct.

The discussion, as well, takes into account all the results in the context of what is known from literature.

I would just suggest, in the first paragraph of the introduction:

  1.  to explain briefly the differences between scenic areas and nature reserves
  2. while talking of the onslaught of the pandemic, to mention the fact that also wild and domestic animals (mammals in particular) have proven to be suscepible to viral infection. There are several reviews about this topic, I know it's not strictly related to your research but I think it should be mentioned when talking about wild animals and the SARS-CoV-2 pandemic.

Author Response

请参阅附件。

Reviewer 2 Report

Comments and Suggestions for Authors

The manuscript titled “Impacts of the COVID-19 pandemic on wildlife in Huangshan Scenic Area, Anhui Province, China." By Lu et al. is important in its area of study- wildlife conservation and public health as it highlights the impact of COVID-19 on the population of wild animals in the study area. However the following concerns need be addressed:

Abstract

The authors overused technical jargon  like "relative abundance indices," "occupancy and detection rates," and "Kernel density curves" which may be difficult for a general scientific audience. They should use more accessible language where possible. E.g., "Kernel density curves showed" can be replaced with "Analysis showed that..."  They also  reiterated that human activity reduction had an impact multiple times in different ways. Overlapping ideas about wildlife abundance changes could be merged to remove redundancy. There was lack of numerical representation of key results, e.g  Statements such as "significantly decreased" and "increased" should include specific statistical values. Add p-values or percentages instead of saying "significantly decreased,"

Introduction

44–73: Redundant because the impact of human activity on wildlife is explained multiple times. Information about human-wildlife interactions should be consolidated

The introduction presents a background but does not clearly state why Huangshan Scenic Area is an ideal study site

62-73: There was overgeneralization of the Anthropause concept because general global trends was discussed but lacking in clear linkage to the study area, thus it did not explicitly state what the study aims to prove

Methods

117–129: This statement “cameras were placed at "human-accessible water sources or animal trails,"  is too broad and vague. How sites were selected to ensure a representative sample?

127: What is/are the rationale/justification for excluding rodents while  selecting only certain species

135–146: Why the choice of Wilcoxon Signed-Rank test, which is a non-parametric test used.

Results

Table 1 and Figure 3 present raw data without summarizing key trends

185–215:  Numerous p-values without synthesizing findings making the  text overloaded with statistics

The authors did not clearly differentiate  expected vs. unexpected findings,  no explanation for why some species showed increased activity while others didnt

Discussion

281–297: Redundant as it repeats trends already presented in result. Focus on why the trends occurred

The ecological consequences of the  observed change in animal activity was  not fully discussed. The authors should discuss briefly how the current study contribute to long-term conservation planning?

There should be consideration for alternative explanations on confounding factors, such as weather conditions or poaching, would these factors have influenced results?

Figures and Tables

Figure 1: Put legend (to indicate elevation or habitat types) for the camera-trap site distribution map

Tables 1–3: Interpretation of key findings immediately is difficult due to excessive numerical data. Authors should group similar species or summarize key findings

Other comments

88–89: Sentence is too long

20–21: Overcomplicated phrasing.

“pandemic” was sometimes used alone, while other times "COVID-19 pandemic" is used.

Although not handled, wild animals were studied and I believe there should be an ethical permission to study the animals

What is/are the limitations of the study

Comments on the Quality of English Language

Minor English editing required

Reviewer 3 Report

Comments and Suggestions for Authors

Line 18 - They were analyzed...     In scientific language, we don´t use personal pronoums

6 => six

Line 20 - These findings…     In scientific language, we don´t use personal pronoums

Keywords: We must avoid to use in keywords, words that are in the title. I suggest:

“Keywords: Wildlife disturbance; Species population trends; Habitat use; Anthropogenic disturbances; Wildlife survival; Coronavirus disease 19.

Line 67: delete “,”

Line 69: delete …”Based on this, we predicted that”…

Line 83: delete…”Consistent with our predictions,”…

Form line 137 to 189: I´ve pointed several times you have used the personal pronoun “We”. We don´t use personal pronoun in scientific language. You must substitute the personal pronoun by other words, passive voice and so on. See the suggestions inside the ballons.

Lines 96-99: these are the objectives, put then in another paragraph, and I suggest to rewrite it:

….”This study objectived to compare the anthropogenic effects before and during the COVID -19 pandemic, in the population size,  habitat use and diurnal activity of seven wild animal species with the highest relative abundance indices (Muntiacus reevesi, Lophura nycthemera, Macaca thibetana, Paguma larvata, Capricornis sumatraensis, Sus scrofa, and Arctonyx collaris), in Huangshan Scenic Area, China.”

Line 307: invert the order: As the mammal and bird species with.....

Line 308: ...the Reeves´muntjac and the silver pheasant, respectivelly, showed…

CONCLUSION:  I SUGGEST THIS:

The reduction in human disturbance during the coronavirus 19 pandemic period allowed most of the seven bird and mammal surveyed species to expand both their population sizes and habitat use, as well as increase their daytime activity. This suggests that anthropogenic disturbances imposed substantial stresses on the ecosystems before the COVID-19 outbreak, with diurnal species being more susceptible to the effects of human activities. This worldwide sanitary crisis has highlighted the connections among humans, nature, and climate change, and brought new scientific knowledge and data that will enable the development of innovative strategies for the coexistence of wildlife and hu
